# Evaluation of Work Mode and Its Importance for Home–Work and Work–Home Relationships: The Role of Resilience, Coping with Stress, and Passion for Work

**DOI:** 10.3390/ijerph192114491

**Published:** 2022-11-04

**Authors:** Ewa Sygit-Kowalkowska, Andrzej Piotrowski, Ole Boe, Samir Rawat, Jelena Minic, Alexandra Predoiu, Radu Predoiu, Žermēna Vazne, Andra Fernate, Romualdas Malinauskas, Nguyen Phuc Nguyen, John Blenkinsopp, Mária Martinská

**Affiliations:** 1Department of Psychology, Kazimierz Wielki University, Leopolda Staffa 1 St., 85-867 Bydgoszcz, Poland; 2Institute of Psychology, University of Gdańsk, Jana Bażyńskiego 4 St., 80-309 Gdańsk, Poland; 3Department of Organisation, Leadership and Management, Inland School of Business and Social Sciences, 2450 Rena, Norway; 4Institute of Psychology, Oslo New University College, 0456 Oslo, Norway; 5Military MIND Academy, Pune 411060, India; 6Faculty of Philosophy, University of Priština, 38220 Mitrovica, Serbia; 7Faculty of Physical Education and Sport, National University of Physical Education and Sports, 060057 Bucharest, Romania; 8Department of Sport and Training Theory, Latvian Academy of Sport Education, Brivibas Gatve 333, LV-1006 Riga, Latvia; 9Department of Physical and Social Education, Lithuanian Sports University, Sporto g. 6, 44221 Kaunas, Lithuania; 10Faculty of Business Administration, University of Economics—The University of Danang, 71 Ngu Hanh Son, Danang City 550000, Vietnam; 11Department of Social Science and Languages, Armed Forces Academy of M.R. Štefánik, Demänová 393, 031 01 Liptovský Mikuláš, Slovakia

**Keywords:** COVID-19, telework, resilience, coping with stress, passion of work, work–home relationship

## Abstract

The COVID-19 pandemic necessitated and facilitated the introduction of telework in organizations. This has also impacted the workers’ relationship between work and private life. The aim of the current study was to examine the links between resilience and mode of work (stationary vs. remote) and the work–home and home–work relationships, and whether they are mediated by passion for work and strategies of coping with stress. The study was carried out on a sample of 1251 participants from Great Britain, India, Latvia, Lithuania, Norway, Poland, Romania, Serbia, Slovakia, and Vietnam. The following measures were used: The Survey Work–Home Interaction, The Brief Resilience Coping Scale, The Passion Scale, and the Brief COPE. Results showed that the more stationary the mode of work, the lower the intensity of the negative influence of personal life on work. Resilience was revealed to have a positive effect on worker functioning. The study also showed a relationship between education and gender and passion for work. Finally, the importance of furthering the knowledge on the home–work and work–home relationships among teleworkers is discussed.

## 1. Introduction

The COVID-19 pandemic necessitated new solutions in social life, including work. To protect employee health and to allow for the continued functioning of organizations, a range of new modes of work was introduced [1]. Implementing remote work has never before progressed at such a pace and scale, despite the fact that part- or full-time remote work is a commonly accepted organization strategy [2]. For obvious reasons, some professions could not transition to remote work (e.g., drivers, medical personnel, uniformed services). Currently available results show that remote work (telework) has not only influenced work itself, but also the work–home relationship [3,4].

Revolutionary changes in the popularity of telework necessitate the search for answers to emerging questions. These concern shaping telework in family-friendly directions, issues of cybersecurity, or eliminating psychological risk factors. Telework is a topic which can be considered from various perspectives: that of the employer, the employee, or the job market itself. The current study focused on the former.


**Remote work (telework) and its importance in fulfilling family roles**


Telework can be defined as “a work practice that involves members of an organization substituting a portion of their typical work hours (ranging from a few hours per week to nearly full-time) to work away from a central workplace—typically from home—using technology to interact with others as needed to conduct work tasks.” [5]. From 2008 to 2019, the percentage of telework employees in the EU aged between 20 and 64 oscillated at around 5% (Eurostat, 2020) [6]. Unplanned, yet rapid and justified by the pandemic, the introduction of telework has led to a large-scale social transformation of work conditions by transferring certain aspects of it to the home environment.

The pandemic has also led to new demands being placed on workers. According to the job demands-resources (JD-R) theory, demands involve effort on the part of the worker to meet them [7,8]. As a consequence, workers accrue emotional, physical, and cognitive costs, which may lead to exhaustion. According to the JD-R theory, as job demands increase, workers use more of their resources to help them to cope with the job demands (characteristics). As defined in the JD-R theory, resources such as competences or organizational support may lower job demands and facilitate task completion and worker development.

Workers’ individual experiences resulting from the implementation of telework are still not described fully in the literature despite a number of empirical investigations published during the pandemic. It is known that telework negates epidemiological risks but, due to social isolation, may cause mental health problems, such as depression [9,10]. It was also shown that limited control over work duration can contribute to excessive demands and workloads placed on workers, which leads to increased stress [11,12]. On the other hand, the ability to work outside of the established hours and outside of the physical workplace gives workers greater autonomy in managing their work and allows them to balance it with personal and family demands [13]. According to Collins et al., telework allows for establishing and developing professional and social relationships with other teleworkers [14]. Telework allows for obtaining appropriate support and managing personal matters more effectively.

The picture revealed through empirical studies is ambiguous, and often contradictory. It is already known that telework employees may experience more negative emotions than stationary workers, which negatively impacts their engagement [15]. This can be a difficult experience both for themselves as well as for their family. Working in the evenings and during weekends intensifies conflicts between various spheres of life, which, in turn, may impact the individual’s satisfaction and productivity [16]. Taken together, the data on the impact of telework on individuals and the knowledge of the mutual influences between the life spheres of work and family lead to the assumption that both a positive as well as a negative relationship may exist there. Thus, the relationships between telework and worker efficiency and health are complex, and require further studies.

Teleworking parents often have to share their workspace, as schools have simultaneously instituted remote teaching. The need to combine telework with childcare in a frequently insufficient physical space and using frequently insufficient equipment has become burdensome for many parents. This is especially the case when both parents work remotely and have more than one child [17,18]. According to Scarpellini et al. over 70% of mothers do not approve of remote teaching due to its demand of significant effort, engagement, and replacing teachers in their roles [19]. When given an option, parents prefer telework, though they also do not approve of situations when their children are simultaneously participating in online learning at home [20]. Studies carried out in the EU show that mothers experience more stress due to telework than do fathers [21]. Teleworking mothers more frequently report pain and discomfort, as well as difficulties in combining work with childcare [22].

In Meijman and Mulder’s effort-recovery model, the environment places demands on an individual, which can be experienced as burdensome. Engaging in coping activities involves effort [23]. Adaptation is made possible by a satisfying regeneration of the organism after effort. A lack of regeneration and the experience of excessive load in one of the individual’s environments may lead to the development of a negative relationship, which will also impact other spheres of functioning.


**Passion for work**


Work carried out with engagement and passion may become a source of satisfaction and fulfillment [24]. Passion can be defined “as a strong inclination toward an activity that people like, that they find important, and in which they invest time and energy.” [25]. Activities which people like and regularly engage in become incorporated into their identity, and the degree of this incorporation depends on the importance of these activities [26]. Passion can be a motivating factor that gives meaning to life. However, it can also have a disorganizing effect by forcing inflexible behavior. In such cases, passion becomes an obstacle to wellbeing and a balanced life. Vallerand argues that two types of passion exist, created through the process of internalization: harmonious and obsessive [27]. In harmonious passion (HP), activity occupies a significant, but not excessive space in the individual’s life compared to other aspects of functioning. Harmonious passion allows for experiencing flow and positive emotions, and it results in higher task orientation.

In obsessive passion (OP), individuals may like their activity, but simultaneously feel pressured to engage in it. In this case, the passion controls the individual, occupying a disproportionally large space in their identity. This leads to conflicts with other forms of activity. Obsessive passion prevents full concentration on tasks and lowers the positive effects of engagement. The pressure to engage in a given activity may generate negative emotions due to the fact that the individual does not engage in other types of activity, for example, family life, to a sufficient degree. The individual becomes distracted, since they cannot separate themselves from their passion. Such activity does not bring satisfaction.

Meta-analyses show that harmonious passion is positively correlated with job results and may increase worker resources, for example, resilience [28]. Defined this way, harmonious passion was considered a resource in the current study. To the best of our knowledge, passion for work has not yet been studied in the context of telework and the work–home relationship.


**Workplace stress in telework and the role of resilience**


The sudden and forced change in mode of work is a difficult and potentially stressful situation. According to Lazarus and Folkman, the creators of the transactional model of stress, a situation has to be appraised as demanding and exceeding the individual’s capabilities in order to be perceived as stressful [29]. The transactional model involves appraisals of the situation through the perspective of coping, which forms the basis of the stress experience. Workers who transitioned into full-time telework had to adapt to a new workplace and meet job demands using new technology. High demands due to mode change together with low control over the situation typically lead to anxiety and exhaustion in workers [30].

Thus far, telework during the pandemic has been variously described as a factor contributing to worker stress. Frequent telework has been shown to both increase [31] as well as decrease stress [32]. Higher autonomy in managing telework is indicated as significantly lowering the feeling of excessive load and workplace stress [33]. Analyses on a sample of workers who were given the option of part-time telework show that they reported lower stress during the telework days [34]. Studies also show that using digital technology at work seems to generate an excessive information flow, leading to higher stress among teleworkers [35]. Telework is related to a reduction in stressful reactions. However, at five days per week, it is related to lower work effectiveness [36].

Coping with a new, stressful situation requires activating personal resources such as resilience [37,38]. Resilience is commonly defined as a relatively constant disposition determining the process of flexible adaptation to constantly changing life demands [32,39]. Studies have confirmed the relationship between resilience and workplace stress [40], well-being [41], and work–life balance [42]. Resilient individuals experience lower stress, report higher wellbeing, and a lower negative impact of work on personal life.


**Aims**


Telework can play a significant role in the home–work and work–home relationships [43]. At the same time, the mutual influence between work and life has long since been proven scientifically. This has particular significance in the context of the popularization of telework, which was mentioned above. Increasing digitalization leads to the assumption that working from home will become increasingly widespread. Thus far, studies on telework during the pandemic mostly concerned individual countries. In order to expand the scientific perspective on telework, a multinational study was carried out. Introducing this heterogeneity into the study sample seemed to be simultaneously novel and universal. It allowed for describing and explaining the degree to which telework affects the home–work and work–home relationship regardless of culture or country. At the same time, emphasis was placed on the participants’ personal characteristics known within the countries cooperating in this project.

The aim of the current study was to examine the links between resilience, work mode, and the work–home and home–work relationships, and whether passion for work and strategies of coping with stress are mediators.

The proposed model is shown in Figure 1.

The following research questions were put forward:What are the participants’ resilience, passion for work, home–work, and work–home relationships, and strategies of coping with stress scores?What is the relationship between the sociodemographic variables, work mode, and resilience and passion for work and strategies of coping with stress?What is the relationship between the sociodemographic variables, work mode, and resilience, and the home–work and work–home relationships?What is the relationship between passion for work and strategies of coping with stress and the home–work and work–home relationships?What is the role of the examined variables (sociodemographic variables, resilience, work mode, strategies of coping with stress, and passion for work) for explaining the home–work and work–home relationships results?

## 2. Materials and Methods


**Study design and participants**


The study was carried out in Great Britain, India, Latvia, Lithuania, Norway, Poland, Romania, Serbia, Slovakia, and Vietnam using Google Forms (Google LLC., Mountain View, CA, USA). Purposive sampling was used. Participation was voluntary, anonymous, and no remuneration was offered. Information about the study was distributed through organizations of potential participants—professionally active individuals. The information about the study contained a general description of its aims and the measures used. It also contained a link to the study. It was highlighted that participants can withdraw their participation at any moment. Participants gave informed consent to participate by marking the appropriate choice in the survey (“I consent” or “I do not consent”) and making a decision to confirm it by clicking the “Send” button. Table 1 shows the sample characteristics. Participants were recruited in a non-random manner. Due to COVID restrictions limiting access to potential participants, snowball sampling was utilized. Additionally, it was also chosen due to its simplicity and relatively low economic demands, which was a significant factor in participant recruitment across the ten countries participating in the project. The aim was to gather as much data as possible. Simultaneously, cross-national comparisons were not assumed a priori.

On 29 June 2021, the study design received approval from the Research Ethics Committee (approval no. 7/15.06.21).

The following measures were used:To measure passion for work, we used the *Passion Scale* [25]. The Passion Scale comprises two subscales for measuring harmonious passion (6 items, e.g., “This work is in harmony with the other activities in my life”) and obsessive passion (6 items, e.g., “I have difficulties controlling my urge to do my work”). The participants rate each item on a 7-point Likert-type scale, from 1 (Not Agree at All) to 7 (Very Strongly Agree). The scale’s reliability, measured by Cronbach’s alpha, ranged from 0.82 (obsessive passion) to 0.86 (harmonious passion).To measure the work–home interaction, the *Survey Work–Home Interaction-Nijmegen* (SWING) Questionnaire by Geurts et al. was used [44]. The scale comprises 22 items measuring four factors: negative and positive work–home interactions (e.g., “Your work schedule makes it difficult for you to fulfil your domestic obligations?; You are better able to keep appointments at home because your job requires this as well?”) and negative and positive home–work interactions (e.g., “You do not fully enjoy your work because you worry about your home situation?; After spending a pleasant weekend with your spouse/family/friends, you have more fun in your job?”). The participants rate each item on a 4-point Likert-type scale, from 1 (never) to 4 (always). The scale’s reliability, measured by Cronbach’s alpha, was 0.76 (for whole scale), 0.72 (for negative work–home interaction) and 0.77 (for positive work–home interaction), 0.73 (for negative home–work interactions) and 0.76 (for positive home–work interactions).*The Brief Resilience Coping Scale* by Smith et al. was used to measure resilience [45]. The scale comprises 6 items (e.g., “I tend to bounce back quickly after hard times.”). The participants rate each item on a 5-point Likert-type scale, from 1 (Definitely disagree) to 5 (definitely agree). The scale’s reliability, measured by Cronbach’s alpha, was 0.83.*Coping Orientation to Problems Experienced Inventory* (Brief-COPE) [46]. This scale comprises 28 items measuring 14 strategies of coping with stress (2 items per coping strategy). The participants indicate how they usually behave in stressful situations. Answers are given on a 4-point Likert-type scale, from 1 (I haven’t been doing this at all) to 4 (I’ve been doing this a lot). The COPE consists of three main groupings, with five subscales per group:
(a)Problem-focused coping: active coping, planning, restraint coping, seeking social support for instrumental reasons and suppression of competing activities;(b)Emotion-focused coping: positive reinterpretation and growth, religion, humor, acceptance, seeking social support for emotional reasons;(c)Dysfunctional coping: focus on and venting of emotions, denial, behavioral disengagement, mental disengagement, and alcohol/drug use.The scale’s reliability, measured by Cronbach’s alpha, ranged from 0.74 (dysfunctional coping), 0.77 (problem-focused coping) to 0.78 (problem-focused coping).


5.The demographic questionnaire containing questions about age, gender, country of residence, and information about the participant’s professional career, shown in Table 1.


The survey also contained an attention check item, intended to test whether the participants are reading the items closely. It had the following form: “This is an attention check which is intended to examine whether the participant is reading the questions. Below, please select “1—I definitely disagree”.


**Data Analysis**


Data analysis was carried out using the IBM SPSS Statistics 27.0 software. First, basic descriptive statistics were calculated together with the Kolmogorov–Smirnov normality test. To establish the relationships between the variables, Pearson’s *r* correlation analysis was carried out for the quantitative variables, Spearman’s *rho* correlation analysis was carried out for the quantitative-ordinal variables, and *Eta* analysis was carried out for the quantitative-nominal variables. Based on the obtained results, IBM AMOS 27.0 was used to create a mediation model of passion for work and strategies of coping with stress in the relationship between sociodemographic variables, resilience, and the home–work and work–home relationships. The following fit criteria were assumed: CFI > 0.95; RMSEA < 0.08; SRMR < 0.08; CMIN/DF < 05. Significance was set at α = 0.05.

To investigate construct validity, the Confirmatory Factor Analysis was proceeded. From the analysis of convergent validity to the theoretical model, seeking to understand when each variable converges to a factor according to the following parametric criteria: average variance extracted (AVE) greater than 0.5; composite reliability (CR) greater than 0.7 and CR >AVE. In addition, maximum shared variance (MSV) and MaxR(H) were used to verify discriminant validity. Finally, a correlation table of the constructs, including the square root of the AVE on the diagonal, was created. Discriminant validity is satisfactory when MSV < AVE and the square root of AVE is greater than inter-construct correlations. For all the analyzed variables (except harmonious passion), AVE was below 0.50, indicating convergent validity problems. Malhotra and Dash note that AVE is a strict measure of convergent validity, so an AVE below 0.50 could be considered if CR is above 0.70 [47,48]. For all constructs, CR was above 0.70, which was acceptable. The MaxR(H) for all constructs was above 0.70, confirming discriminant validity. MSV > AVE held true only for emotional focused strategies and problem focused strategies, indicating a strong correlation between these two constructs.

## 3. Results

### 3.1. Descriptive Statistics

A total of 1251 people took part in the study. Participants were from Great Britain, India, Latvia, Lithuania, Norway, Poland, Romania, Serbia, Slovakia, and Vietnam. Table 1 shows the sample characteristics.

Table 2 shows the analysis of basic descriptive statistics for the quantitative variables. Additionally, the Kolmogorov–Smirnov test was used to test the normality of their distribution. The analysis showed that neither of the analyzed variables assumed a normal distribution. Nevertheless, considering the sample size and the skewness values in the −1;1 range, it can be assumed that the non-normality of the distribution was not significant.

### 3.2. Relationships between the Sociodemographic Variables, Work Mode, Resilience, and Passion for Work, and Strategies of Coping with Stress

First, the relationships between the sociodemographic variables, work mode, resilience, and passion for work and coping with stress were examined. Gender was positively correlated with obsessive passion, harmonious passion, and the three coping strategy groups. Higher passion for work and more frequent use of all coping strategy groups were reported by women. Age and total work experience were positively, though weakly correlated with both dimensions of passion for work, and negatively correlated with dysfunctional coping strategies—the older the participants and the longer their work experience, the higher their obsessive and harmonious passion and the less frequent their use of dysfunctional coping strategies. Education was weakly and negatively correlated with obsessive passion and dysfunctional coping strategies. The higher the participants’ education level, the lower their obsessive passion and the less frequent their dysfunctional coping strategy use. Resilience was negatively correlated with obsessive passion (weak correlation) and dysfunctional coping strategies (moderate correlation), and positively and weakly correlated with harmonious passion and emotion-focused coping strategies. Harmonious passion was higher and emotion-focused coping strategy use was more frequent at higher resilience levels, while obsessive passion was lower and dysfunctional coping strategy use was less frequent. The results are shown in Table 3.

### 3.3. Relationships between Passion for Work and Strategies of Coping with Stress and the Home–Work and Work–Home Relationships

The analysis also showed a positive link between passion for work and strategies of coping with stress and positive WHI and HWI, as well as negative HWI. Moreover, obsessive passion and dysfunctional coping strategies were positively, and harmonious passion negatively, correlated with negative WHI. The correlations were weak to moderate in size. Higher frequencies of using all groups of strategies of coping with stress and higher passion of work was reported at higher levels of positive WHI and HWI as well as negative HWI. Higher levels of negative WHI were reported at higher levels of obsessive passion and more frequent dysfunctional coping strategy use. The results are shown in Table 4.

### 3.4. Model

Based on the correlation analysis, a path analysis was carried out. Work mode was excluded from the analysis due to the fact that it was not related to the mediating variables, and from among the explained variables, it was only related to negative HWI. The model included the covariance between resilience and gender as well as between total work experience, education, and age. Moreover, the model included the relationship between passion for work and strategies of coping with stress. The analyzed model showed a good fit to data, χ^2^(34) = 132.34; *p* < 0.001; CMIN/DF = 3.89; CFI = 0.958; SRMR = 0.043; RMSEA = 0.049 [90% CI: 0.041; 0.058].

Table 5 shows the regression coefficients for the paths included in the model. Gender was statistically significantly related to obsessive passion, harmonious passion, and all three groups of coping strategies. A statistically significant, positive relationship was also observed for positive HWI, which indicates higher levels of this variable in women. Age was related to using dysfunctional coping strategies—the older the participants, the lower their frequency of using dysfunctional coping strategies. Education was negatively related to obsessive passion and dysfunctional coping strategy use, and positively related to negative WHI. At higher education levels, obsessive passion and dysfunctional coping strategy use were lower, while negative WHI was higher. Resilience was negatively related to obsessive passion, dysfunctional coping strategy use, and negative WHI, and positively related to harmonious passion and emotion-focused coping strategy use. Total work experience was positively related to obsessive and harmonious passion, as well as to dysfunctional coping strategy use. Harmonious passion was positively related to positive HWI and WHI, and negatively related to negative WHI. Obsessive passion was positively related to all aspects of the home–work and work–home relationships. Emotion-focused coping strategy use was negatively related to negative HWI. Problem-focused coping strategy use was positively related to positive HWI and WHI, whereas dysfunctional coping strategy use was positively related to positive WHI as well as negative WHI and HWI.

To examine whether passion for work and strategies of coping with stress had a significant mediating role in the sociodemographic variables, resilience, and the home–work and work–home relationships, an additional analysis was carried out using bootstrapping with 5000 samples.

Table 6 shows the total, direct, and indirect effects between the independent and dependent variables. The analysis showed statistically significant indirect effects of education on positive WHI and negative WHI and HWI, of gender and total work experience on positive WHI and positive and negative HWI, of resilience on negative WHI, and of age on positive WHI and negative HWI.

To establish which of the passion for work factors and groups of coping strategies were mediators, additional analyses for individual paths were carried out (see Table 7).

Female gender increased obsessive passion and simultaneously increased positive and negative HWI and WHI. Through increases in harmonious passion, positive HWI and WHI also increased, while negative WHI decreased. Through increases in emotion-focused coping strategy use, negative HWI increased. On the other hand, through increases in problem-focused coping strategy use, positive HWI increased. In women, a higher frequency of dysfunctional coping strategy use increased positive WHI as well as negative WHI and HWI.

Age, through increases in dysfunctional coping strategy use, lowered positive WHI as well as negative WHI and HWI.

Education negatively influenced obsessive passion and dysfunctional coping strategy use, simultaneously lowering positive and negative WHI and negative HWI. Through lowering obsessive passion, education also lowered positive HWI.

Resilience, through increases in harmonious passion, increased positive HWI and WHI and lowered negative WHI. Through increases in emotion-focused coping strategy use, resilience also increased negative HWI. Resilience negatively influenced dysfunctional coping strategy use, simultaneously lowering positive WHI as well as negative WHI and HWI.

Total work experience, through obsessive passion, increased all dimensions of passion for work, and through harmonious passion, increased positive HWI and WHI. On the other hand, it lowered negative WHI. Total work experience also increased negative WHI and HWI through dysfunctional coping strategy use.

## 4. Discussion

Every crisis, including the COVID-19 pandemic, necessitates the implementation of new solutions or facilitates a broader adoption of heretofore niche ones. The necessity to protect worker health while maintaining the continued functioning of organizations has led to an increase in the proportion of telework compared to stationary work [49]. Rapid and mass changes in the mode of work are related to challenges for both the organizations as well as their employees and their families. Telework gives workers more opportunities to work during their preferred times. On the other hand, it may blur the lines between private life and work.

The current study focused on the relationships between worker characteristics, mode of work, and the interactions between work and home which arose as a result of governmental restrictions during the pandemic and for many are now a permanent feature of their working lives. Previous studies on this topic have yielded a complex picture of worker experiences in light of the widespread implementation of telework. Analyzing this issue remains pertinent and can be useful for planning organizational solutions in the future.

The authors of the SWING questionnaire, used in the current study to measure the explained variable of the work–home and home–work relationships, define this interaction as “a process in which a worker’s functioning (behavior) in one domain (e.g., home) is influenced by (negative or positive) load reactions that have built up in the other domain (e.g., work” (p. 322) [44]. During the pandemic, the mode of work may have changed to part- or full-time telework, which was considered in the current study as a potentially significant factor in the home–work interaction. In the pandemic context, a climate of trust and space for independence in the employee group has also acquired a different meaning, which is highlighted in the literature as a factor shaping individual resilience [50].

The current study shows the positive role of resilience, which is consistent with previous results on its importance in individual functioning. Resilience is indicated as a significant factor in mental health [51,52] which facilitates flexibility, adaptation, and employee engagement [53,54]. In the current study, higher resilience co-occurred with higher harmonious passion and more frequent emotion-focused coping strategy use, as well as with lower obsessive passion and less frequent dysfunctional coping strategy use. Dysfunctional coping strategies involve, among others, denial, behavioral disengagement, mental disengagement, and alcohol/drug use. Workers exhibiting such tendencies disengage from purposeful activity oriented at managing difficult situations in factor of activities defending against the difficulties of reality. This is particularly important for organizations, as research shows that dysfunctional coping is negatively associated with work performance [55]. Prior studies show that obsessive passion for work is positively related with maladaptive coping strategies such as catastrophizing or rumination, which lead to higher emotional suffering [56]. Obsessive passion leads to a state of addiction to activity [57]. This is consistent with the strategy of directing attention away from difficult situations through drug use. Moreover, the current study showed that resilience, through increasing harmonious passion, increases positive HWI and WHI, and lowers negative WHI. This result confirms the protective role of resilience [58].

Prior studies did not indicate the existence of gender differences in terms of either of the passion dimensions [59]. However, the current results show that women are more likely to have obsessive passion for work and thus intensifies the influence of positive and negative HWI and WHI. Thus far, obsessive passion has been indicated as a factor which correlates more strongly with difficulties in balancing work and home roles, in contrast to harmonious passion [60]. The current results are significant in the context of the strong and positive correlation between harmonious passion and job satisfaction [61].

An individual’s relationships between their professional and family roles may take the form of a negative influence (pressure) of one environment on another. This is described as a conflict, where a contradiction and incompatibility regarding the ability to meet the demands of both of the roles emerge [62]. A lack of balance between the professional and private sphere may lead to poorer health and role conflict in women. However, the overall picture of the costs and benefits of work for women remains unclear despite the ability to introduce such work–home practices as telework [63,64]. Studies on the tendency by women to experience obsessive passion are important also due to the fact that obsessive passion is positively correlated with lower levels of positive affect, also as a result of unfulfilled personal needs [65,66]. The pandemic period during which data collection took place was especially difficult for working mothers. Women on the job market reported experiencing overload due to having to care for small children staying at home and additional, unpaid care for other family members. It has been reported that teleworking mothers experience higher negative WHI as a result of simultaneously fulfilling the worker and carer roles [67]. On the other hand, surprisingly, studies also showed a higher positive influence of work on private life and of private life on work among women. It would be interesting to examine whether the experience of a positive home–work/work–home influence is the result of conscious activity in this direction or whether it is a consequence of received or perceived social support in these areas.

One result was particularly interesting for the discussion of work performance and meaning for employee functioning. Mode of work was revealed to be negatively and weakly correlated with negative HWI, meaning that the more stationary the mode of work, the lower the intensity of negative influence of personal life on work. Its further significance was ruled out in the model-based analyses. Thus, the current results converge with the studies showing that home–work relationships involve pressure to fulfill both these roles [68].

Finally, it is worth noting that the current analyses showed the significance of education. At higher education levels, obsessive passion and dysfunctional coping strategy use were lower, while the intensity of negative WHI was higher. This may indicate a positive role of access to education and opportunities for knowledge building, thus confirming the hypothesis that education is vital for a satisfying life [69]. Education grants experience and skill development, which contribute to personal development.

## 5. Strengths, Limitations, and Future Research

The study presented above was multinational in nature. It presents a more comprehensive picture of the influence of mode of work on the home–work and work–home relationship. Existing scientific analyses of telework were mostly based on single-country studies.

Moreover, the practical value of the current study is that its results can be used by representatives of organizations from various countries. The issues examined in the current study, such as resilience, coping with stress, and passion for work are established psychological constructs. They can be shaped during personal development, which suggests a real possibility of utilizing the current results. Knowledge on competence development in these areas is broadly accessible, which allows for seeking solutions for optimal work development. The study also concerns the seemingly well-researched area of gender at work. Simultaneously, they shed light on the role of passion for work, which differentiated the current sample. This represents an additional voice in the analyses of workplace issues from the point of view of the employee, or the individual directly impacted by telework solutions.

Although the current study was carried out on a large, multinational sample, it requires continuation. Controlling the workplace characteristics as a variable in the research model would be significant. During the pandemic, certain aspects related to the responsibilities of the state became subjects of popular discussion: education of children, adolescents, and adults, the development of public telemedicine, or public administrative services for citizens. Considering the specifics of particular jobs seems very significant for further studies.

A publication by the World Health Organization indicates that protecting worker health during the pandemic will improve their ability to fulfill their roles [70]. The interrelationships between the professional and the private spheres, also taking into account the mode of work (telework, stationary work), leads to the question of which solutions are beneficial for organizations and workers, and which are not. Prior studies show that the issue of time devoted to work requires further analyses which would consider what role fulfillment entails for the workers. Teleworkers spend additional time on work [71]. Studies show that email or phone contact outside of work hours is related to lower psychophysiological wellbeing in workers [72]. At the same time, saving time due to having eliminated commuting is indicated as a benefit of telework [73].

Although our goal was not to study the issue of family life-work balance, or managing both these areas of life, we did consider the problem of one area influencing the other. However, a more complex approach to this topic would be interesting. Role (home–work) conflict is positively related to experiencing difficult emotions in various areas of life [74]. Difficulties in reconciling various roles are a significant factor shaping a range of negative outcomes in workers. Thus, it is worthwhile to examine the consequences of both positive and negative influences of home life on work, and vice versa. Employee health seems important to explore in further studies, both in the context of exhibited health behaviors as well as of determining heath characteristics, including mental health. Mode of work may impact the frequency of physical activity and dietary choices, in turn influencing quality of life. Studies on coping with stress and on health also lead to considering psychoactive substance use during telework. Dyad-based studies appear equally important. They would allow for collecting data from the participants’ life partners on the mutual influences between telework and private life. The phenomenon of telework may also be considered from the point of view of gains and losses. Based on Adams’ equity theory, motivation to work in a different mode may change when the effort is not appropriately rewarded [75]. This may be important to consider when designing new study models.

Further analyses of the home–work and work–home relationships seem particularly important in women. Nevertheless, studies on telework should also consider social role fulfillment by working fathers. Existing analyses show that their engagement at home may change depending on the situation. However, the culturally determined traditional masculine social role is also important [76,77].

Optimal need fulfillment in the personal and professional spheres are becoming a source of personal satisfaction, but they may assume different forms in the context of widespread telework. The literature indicates that practical solutions in terms of eliminating or reducing roles as well as sharing them with a life partner are a significant factor determining women’s achievements [78]. In light of the experienced difficulties, individuals may make specific decisions related to parenthood. Due to its sensitive nature, studies of this issue could take an interview-based form. In an interview-based study, Crosbie and Moore showed that teleworkers develop numerous strategies of coping with everyday isolation and stress, though the increased number of work hours negatively impact their family life [79].

Finally, to further the knowledge on the current challenges facing professionally active people, it would be worthwhile to continue studies on resilience in telework. Regarding such professional groups as nurses, who do not work remotely, studies show that building resilience is possible through, among others, developing positive workplace relationships [80]. It would be interesting to examine which activities would serve as appropriate substitutes in samples of teleworkers.

## 6. Conclusions

Based on the analyses of the current results as well as on the available literature, it can be concluded that:-The results are consistent with a positive picture of the role of resilience in individual functioning, including work, through the regulation of passion for work and the home–work and work–home relationships.-Women tend to have higher levels of obsessive passion for work and is thus important for understanding the issue of home–work and work–home relationships.-The more stationary the mode of work, the lower the negative influence of personal life on work, which is particularly pertinent to the context of the current study.-The current study shows a relationship between education and passion for work, the tendency to engage in specific coping strategies, and the influence of work on family life.

## Figures and Tables

**Figure 1 ijerph-19-14491-f001:**
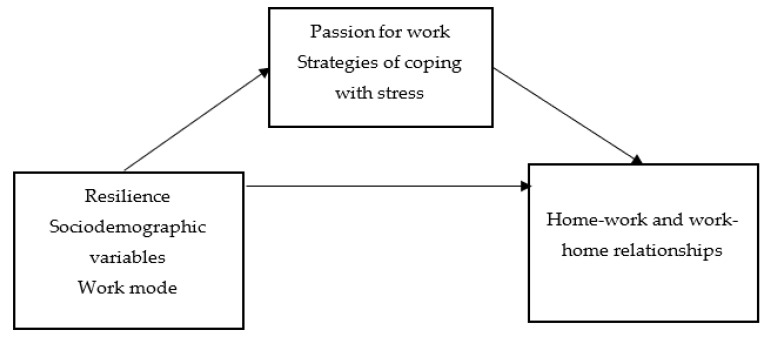
Proposed research model.

**Table 1 ijerph-19-14491-t001:** Descriptive statistics of surveyed respondents.

Variable	Value
**Gender, *n(%)***	
Man	456 (36.5)
Woman	788 (63.0)
Missing	7 (0.6)
**Age, *M(SD)***	35.59 (11.91)
**Country, *n(%)***	
Poland	417 (33.3)
Vietnam	47 (3.8)
Slovakia	24 (1.9)
Serbia	182 (14.5)
Romania	131 (10.5)
Lithuania	212 (16.9)
Latvia	196 (15.7)
India	18 (1.4)
Great Britain	19 (1.5)
Norway	4 (0.3)
Missing	1 (0.1)
**Total work experience, *M(SD)***	13.76 (11.64)
**Work experience in current position, *M(SD)***	8.50 (9.08)
**Education, *n(%)***	
Vocational education	43 (3.4)
Secondary education	321 (25.7)
Bachelor’s/engineering degree	393 (31.4)
Master’s	479 (38.3)
Missing	15 (1.2)
**Currently continuing education, *n(%)***	
Yes, in full-time studies	239 (19.1)
Yes, in the evening studies	36 (2.9)
Yes, in extramural studies	221 (17.7)
No	751 (60.0)
Missing	4 (0.3)
**Position held, *n(%)***	
Serial/executive	235 (18.8)
Specialized	516 (41.2)
Managerial	159 (12.7)
Director	119 (9.5)
Business owner	196 (15.7)
Missing	26 (2.1)
**Form of employment, *n(%)***	
Employment contract for a trial period	51 (4.1)
Fixed-term employment contract (including replacement contract)	255 (20.4)
Employment contract for an indefinite period	663 (53.0)
Contract of mandate	55 (4.4)
Contract work	5 (3.2)
An agency agreement	5 (0.4)
A contract for the duration of a specific job	22 (1.8)
Own business	99 (7.9)
Others	57 (4.6)
Missing	4 (0.3)
**Employment Industry, *n(%)***	
Public administration	116 (9.3)
Trade	103 (8.2)
Services	177 (14.1)
Finance and banking	53 (4.2)
Tourism	60 (4.8)
Education	357 (28.5)
Medical care	97 (7.8)
Uniformed services	59 (4.7)
Others	222 (17.7)
Missing	7 (0.6)
**Stationary, remote or mixed mode of work, *n(%)***	
Completely remotely	189 (15.1)
In a mixed system	482 (38.5)
Completely stationary	574 (45.9)
Missing	6 (0.5)
**Percentage remotely, *n(%)***	
Up to 30%	154 (12.3)
From 30% to 60%	221 (17.7)
Over 60%	195 (15.6)
Not applicable—I work stationary	556 (44.4)
Missing	125 (10.0)
**Since when remotely, *n(%)***	
Up to 3 months	78 (6.2)
From 3 to 6 months	224 (17.9)
Over 6 months	337 (26.9)
Not applicable—I work stationary	545 (43.6)
Missing	67 (5.4)
**Space in the house for remote work, *n(%)***	
Yes	569 (45.5)
No	156 (12.5)
Not applicable—I work stationary	517 (41.3)
Missing	9 (0.7)
**Computer equipment for remote work, *n(%)***	
Yes	678 (54.2)
No	74 (5.9)
Not applicable—I work stationary	494 (39.5)
Missing	5 (0.4)
**Combine teleworking with childcare or similar, *n(%)***	
Yes	269 (21.5)
No	453 (36.2)
Not applicable—I work stationary	522 (41.7)
Missing	7 (0.6)
**Work remotely before COVID, *n(%)***	
Yes	146 (11.7)
No	1093 (87.4)
Missing	12 (0.9)

**Table 2 ijerph-19-14491-t002:** Descriptive statistics together with the Kolmogorov–Smirnov test of normality of distribution (N = 1251).

Variable	*M*	*Me*	*SD*	*Sk.*	*Kurt.*	*Min.*	*Max.*	*D*	*p*
**Passion for work**									
Obsessive Passion	2.73	2.50	1.14	0.56	−0.20	1.00	6.83	0.08	<0.001
Harmonious Passion	4.64	4.67	1.19	−0.31	−0.27	1.00	7.00	0.06	<0.001
**Resilience**	3.24	3.33	0.75	−0.31	−0.39	1.00	5.00	0.10	<0.001
**Home–work and work–home relationships**									
Negative WHI	1.87	1.88	0.48	0.75	0.99	1.00	4.00	0.13	<0.001
Positive WHI	2.23	2.17	0.47	0.27	0.14	1.00	4.00	0.12	<0.001
Negative HWI	1.91	1.75	0.61	0.55	0.06	1.00	4.00	0.12	<0.001
Positive HWI	2.42	2.33	0.55	0.32	0.05	1.00	4.00	0.09	<0.001
**Strategies of coping with stress**									
Emotion–focused strategies	2.54	2.50	0.48	−0.10	0.63	1.00	4.00	0.05	<0.001
Problem–focused strategies	2.94	3.00	0.56	−0.60	0.73	1.00	4.00	0.11	<0.001
Dysfunctional coping strategies	2.06	2.00	0.43	0.32	0.34	1.00	3.58	0.07	<0.001

M—mean; Me—median; SD—standard deviation; Sk.—skewness; Kurt.—kurtosis; Min.—minimal value; Max.—maximal value; D—Kolmogorov–Smirnov test statistic; *p*—statistical significance; WHI—work–home interaction; HWI—home–work interaction.

**Table 3 ijerph-19-14491-t003:** Correlations between the sociodemographic variables, work mode, and resilience and passion for work and strategies of coping with stress.

Variable	Obsessive Passion	Harmonious Passion	Emotion-Focused Strategies	Problem-Focused Strategies	Dysfunctional Coping Strategies
Gender	0.08 **	0.06 *	0.11 **	0.06 *	0.16 **
Age	0.16 **	0.09 **	−0.03	−0.05	−0.09 **
Total work experience	0.19 **	0.12 **	−0.01	−0.04	−0.06 *
Education	−0.08 **	−0.01	0.01	0.03	−0.13 **
Work mode: stationary, remote, or mixed	−0.03	−0.04	−0.01	0.02	−0.02
Resilience	−0.07 *	0.19 **	0.11 **	0.05	−0.36 **

* *p* < 0.05; ** *p* < 0.01.

**Table 4 ijerph-19-14491-t004:** Correlations between passion for work and strategies of coping with stress and the home–work and work–home relationships.

Variable	Negative WHI	Positive WHI	Negative HWI	Positive HWI
Obsessive Passion	0.26 **	0.24 **	0.27 **	0.27 **
Harmonious Passion	−0.25 **	0.27 **	0.15 **	0.39 **
Emotion focused strategies	0.01	0.19 **	0.18 **	0.27 **
Problem focused strategies	0.05	0.24 **	0.09 **	0.28 **
Dysfunctional coping strategies	0.30 **	0.16 **	0.22 **	0.11 **

* *p* < 0.05; ** *p* < 0.01; WHI—work–home interaction; HWI—home–work interaction.

**Table 5 ijerph-19-14491-t005:** Regression coefficients for the path model.

X	Y	B	SE	CR	*p*	β
Gender	Obsessive Passion	0.20	0.07	2.90	0.004	0.08
Gender	Harmonious Passion	0.26	0.07	3.73	<0.001	0.11
Gender	Emotion–focused strategies	0.14	0.03	5.28	<0.001	0.14
Gender	Problem–focused strategies	0.08	0.03	2.33	0.020	0.07
Gender	Dysfunctional coping strategies	0.10	0.02	4.03	<0.001	0.11
Age	Dysfunctional coping strategies	−0.01	0.00	−3.95	<0.001	−0.29
Education	Obsessive Passion	−0.10	0.04	−2.87	0.004	−0.08
Education	Dysfunctional coping strategies	−0.05	0.01	−3.78	<0.001	−0.10
Resilience	Obsessive Passion	−0.10	0.04	−2.23	0.026	−0.07
Resilience	Harmonious Passion	0.29	0.04	6.83	<0.001	0.19
Resilience	Emotion–focused strategies	0.05	0.01	3.18	0.001	0.07
Resilience	Dysfunctional coping strategies	−0.20	0.02	−13.33	<0.001	−0.36
Total work experience	Obsessive Passion	0.04	0.01	4.54	<0.001	0.39
Total work experience	Harmonious Passion	0.03	0.01	3.00	0.003	0.25
Total work experience	Dysfunctional coping strategies	0.01	0.00	3.27	0.001	0.24
Gender	Positive HWI	0.08	0.03	2.65	0.008	0.07
Education	Negative WHI	0.08	0.01	5.60	<0.001	0.14
Resilience	Negative WHI	−0.06	0.02	−3.66	<0.001	−0.10
Harmonious Passion	Positive HWI	0.12	0.02	7.96	<0.001	0.25
Harmonious Passion	Positive WHI	0.06	0.01	4.59	<0.001	0.16
Harmonious Passion	Negative WHI	−0.17	0.01	−13.19	<0.001	−0.40
Obsessive Passion	Positive HWI	0.07	0.01	4.93	<0.001	0.15
Obsessive Passion	Positive WHI	0.07	0.01	5.67	<0.001	0.16
Obsessive Passion	Negative WHI	0.16	0.01	13.96	<0.001	0.38
Obsessive Passion	Negative HWI	0.10	0.02	6.17	<0.001	0.19
Emotion–focused strategies	Negative HWI	0.19	0.05	3.99	<0.001	0.15
Problem–focused strategies	Positive WHI	0.12	0.03	3.78	<0.001	0.14
Problem–focused strategies	Positive HWI	0.16	0.04	4.34	<0.001	0.16
dysfunctional coping strategies	Positive WHI	0.07	0.03	2.21	0.027	0.07
dysfunctional coping strategies	Negative WHI	0.17	0.03	5.32	<0.001	0.15
dysfunctional coping strategies	Negative HWI	0.17	0.04	3.83	<0.001	0.12

B—unstandardized regression coefficient, SE—standard error; CR—critical ratio; *p*—statistical significance; β—standardized regression coefficient. Note: Only statistically significant results are shown in the table.

**Table 6 ijerph-19-14491-t006:** Summary of total, direct, and indirect effects in the explanatory model of the home–work and work–home relationships—standardized regression coefficients with a 95% CI.

		Education	Gender	Resilience	Total Work Experience	Age
**Total effect**	Positive WHI	**−0.02** **[−0.03; −0.01]**	0.06 [−0.01; 0.12]	0.00 [−0.04; 0.03]	**0.12** **[0.07; 0.19]**	**−0.06** **[−0.13; −0.01]**
	Positive HWI	−0.01 [−0.02; 0.06]	**0.12** **[0.06; 0.18]**	**0.07** **[0.01; 0.12]**	**0.14** **[0.06; 0.23]**	−0.05 [−0.13; 0.01]
	Negative WHI	**0.10** **[0.03; 0.16]**	0.01 [−0.03; 0.03]	**−0.25** **[−0.31; −0.19]**	0.08 [−0.01; 0.18]	−0.06 [−0.15; 0.04]
	Negative HWI	**−0.03** **[−0.04; −0.01]**	**0.05** **[0.03; 0.07]**	−0.05 [−0.12; 0.01]	**0.11** **[0.06; 0.17]**	**−0.06** **[−0.12; −0.02]**
**Direct effect**	Positive WHI	–	0.01 [−0.05; 0.06]	–	–	–
	Positive HWI	–	**0.07** **[0.01; 0.12]**	0.03 [−0.03; 0.08]	0.02 [−0.03; 0.07]	–
	Negative WHI	**0.14** **[0.08; 0.20]**	–	**−0.10** **[−0.16; −0.04]**	–	–
	Negative HWI	–	–	−0.02 [−0.08; 0.05]	–	–
**Indirect effect**	Positive WHI	**−0.02** **[−0.03; −0.01]**	**0.05** **[0.02; 0.08]**	0.00 [−0.04; 0.03]	**0.12** **[0.07; 0.19]**	**−0.06** **[−0.13; −0.01]**
	Positive HWI	−0.01 [−0.02; 0.01]	**0.06** **[0.03; 0.09]**	0.04 [0.01; 0.08]	**0.12** **[0.06; 0.20]**	−0.05 [ −0.13; 0.01]
	Negative WHI	**−0.04** **[−0.07; −0.02]**	0.01 [−0.03; 0.03]	**−0.15** **[−0.19; −0.11]**	0.08 [−0.01; 0.18]	−0.06 [−0.15; 0.04]
	Negative HWI	**−0.03** **[−0.04; −0.01]**	**0.05** **[0.03; 0.07]**	−0.04 [−0.07; 0.01]	**0.11** **[0.06; 0.17]**	**−0.06** **[−0.12; −0.02]**

WHI—work–home interaction; HWI—home–work interaction. Note: Statistically significant correlations are given in bold.

**Table 7 ijerph-19-14491-t007:** Indirect effect coefficients for individual paths—specific effects.

Indirect Path	*B*	*LL*	*UL*	*p*	*β*
Gender → Obsessive Passion → Positive HWI	0.01	0.01	0.03	0.007	0.01
Gender → Obsessive Passion → Positive WHI	0.01	0.01	0.02	0.009	0.01
Gender → Obsessive Passion → Negative WHI	0.03	0.01	0.05	0.011	0.03
Gender → Obsessive Passion → Negative HWI	0.02	0.01	0.04	0.009	0.01
Gender → Harmonious Passion → Positive HWI	0.03	0.02	0.05	0.002	0.03
Gender → Harmonious Passion → Positive WHI	0.02	0.01	0.03	0.001	0.02
Gender → Harmonious Passion → Negative WHI	−0.04	−0.07	−0.02	0.001	−0.04
Gender → Emotion–focused strategies → Negative HWI	0.03	0.01	0.04	0.001	0.02
Gender → Problem–focused strategies → Positive HWI	0.01	0.00	0.02	0.048	0.01
Gender → Dysfunctional coping strategies → Positive WHI	0.01	0.00	0.02	0.034	0.01
Gender → Dysfunctional coping strategies → Negative WHI	0.02	0.01	0.03	0.001	0.02
Gender → Dysfunctional coping strategies → Negative HWI	0.02	0.01	0.03	0.001	0.01
Age → Dysfunctional coping strategies → Positive WHI	<−0.01	0.00	0.00	0.045	−0.02
Age → Dysfunctional coping strategies → Negative WHI	<−0.01	0.00	0.00	0.002	−0.04
Age → Dysfunctional coping strategies → Negative HWI	<−0.01	0.00	0.00	0.003	−0.03
Education → Obsessive Passion → Positive HWI	−0.01	−0.01	0.00	0.010	−0.01
Education → Obsessive Passion → Positive WHI	−0.01	−0.01	0.00	0.009	−0.01
Education → Obsessive Passion → Negative WHI	−0.02	−0.03	−0.01	0.013	−0.03
Education → Obsessive Passion → Negative HWI	−0.01	−0.02	0.00	0.012	−0.01
Education → Dysfunctional coping strategies → Positive WHI	−0.01	−0.01	0.00	0.017	−0.01
Education → Dysfunctional coping strategies → Negative WHI	−0.01	−0.02	0.00	0.000	−0.01
Education → Dysfunctional coping strategies → Negative HWI	−0.01	−0.02	0.00	0.000	−0.01
Resilience → Harmonious Passion → Positive HWI	0.03	0.02	0.05	0.001	0.05
Resilience → Harmonious Passion → Positive WHI	0.02	0.01	0.03	0.001	0.03
Resilience → Harmonious Passion → Negative WHI	−0.05	−0.07	−0.03	0.001	−0.08
Resilience → Emotion–focused strategies → Negative HWI	0.01	0.00	0.02	0.004	0.01
Resilience → Dysfunctional coping strategies → Positive WHI	−0.02	−0.03	0.00	0.031	−0.02
Resilience → Dysfunctional coping strategies → Negative WHI	−0.03	−0.05	−0.02	0.000	−0.05
Resilience → Dysfunctional coping strategies → Negative HWI	−0.03	−0.05	−0.02	0.001	−0.04
Total work experience → Obsessive Passion → Positive HWI	<0.01	0.00	0.00	0.001	0.06
Total work experience → Obsessive Passion → Positive WHI	<0.01	0.00	0.00	0.001	0.06
Total work experience → Obsessive Passion → Negative WHI	0.01	0.00	0.01	0.003	0.15
Total work experience → Obsessive Passion → Negative HWI	<0.01	0.00	0.01	0.002	0.07
Total work experience → Harmonious Passion → Positive HWI	<0.01	0.00	0.01	0.017	0.06
Total work experience → Harmonious Passion → Positive WHI	<0.01	0.00	0.00	0.014	0.04
Total work experience → Harmonious Passion → Negative WHI	<−0.01	−0.01	0.00	0.015	−0.01
Total work experience → Dysfunctional coping strategies → Negative WHI	<0.01	0.00	0.00	0.004	0.04
Total work experience → Dysfunctional coping strategies → Negative HWI	<0.01	0.00	0.00	0.005	0.03

B—unstandardized regression coefficient for indirect effects; LL and UL—lower and upper limit of the 95% confidence interval; *p*—statistical significance; β—standardized regression coefficient for indirect effects; WHI—work–home interaction; HWI—home–work interaction. Note: Only statistically significant results of the path analysis are shown.

## Data Availability

The data presented in this study are available on request from the corresponding author.

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
