# Peer review of "Evaluation of Work Mode and Its Importance for Home–Work and Work–Home Relationships: The Role of Resilience, Coping with Stress, and Passion for Work"

_ijerph, 2022, doi:10.3390/ijerph192114491_

Round 1

Reviewer 1 Report

Dear Authors

Thank you so much for submitting your paper in this outlet "International Journal of Environmental Research and Public Health".

I read your paper and mentioned my concern down here:

1. The title seems too stuffy. Would you make it shortened?

2. Introduction section lacks rigor and depth. What is the problem statement? How is the paper original? What is its contribution can not be generated from introduction section.
3. Additionally, why does this paper deserves attention from readers when the pandemic is about to over. 

4. The citations used in this paper are outdated. It needs recent citations particularly from 2020-2022. The authors have used few from 2021 but I found no paper from 2022.

5. The authors used data from multi-countries. Surprisingly, those are not representative. Particularly, responses from India, Britain, and Norway are too few to mention. I will insist to increase it so that it represent countries like India, Britain, and Norway.

6. Authors did not mention convergent validity issues! Did those data converge to their latent variables?

7. What about the method bias issues? Are they bias free?

8. Does the estimates mentioned in Table 3 really demonstration correlation or regression? It seems regression? It requires clarification.

9. The strengths of the paper mentioned in section 6 must be supported and justified. These strengths must be further improved.

Wish you all the best.

Author Response

Dear Professor

Thank you for reviewing our article, it allowed us to improve the quality of our work.

We have decided to submit all reviews with responses so that you can follow the progress of our work.

Regards,

Authors

Reviewer 2 Report

Dear authors, there are still some critical problems in the current form of  manuscript needed to be addressed.

(1) The title of this manuscript could be attractive, which suggests that the article would be a "more reliable" multi-country study. However, the performance of the data collection and analysis fail to meet such expection. Table 1 shows that, only 4 respondents come from Norway, 18 respondents come from India, 19 respondents come from Britain, and disproportionate sample weight is clusterd in Polard (417 respondents). I am not sure how the sample weight is determined. If it is just a convenience sample, and there is no consideration about the sample weight, is it appropriate to "attract" the attention of readers by using such a title?

If the authors indeed would like to conduct a reliable multi-country investigation, the most convincing way is to improve the reserach design and to conduct another one study (either quantitative or qualitative) to make this article a mixed investigation for cross-validation.

(2) The reserach model (Fig 1) shows that the passion for work would affects the home-work and work-home relationship. However, according to the scale, the passion for work is measured by items such as "This work is in harmony with the other activities in my life”, and the work-home interaction is measured by "Your work schedule makes it difficult for you to fulfil your domestic obligations?". There is reasonable doubt about the problem of reverse causality.

(3) The comparison of model of different factor is absent, and there only very brief discussion about this point (line 333-335).

Author Response

(The authors gave the same response as above.)

Reviewer 3 Report

1.     There was a section about remote work (telework) and its importance in fulfilling family roles. I do believe that telework may adversely affect some workers’ family roles. Please state some negative impacts of remote work on the family domains. Also, how about the importance and adverse effects of remote work on work roles? I think these should be clearly demonstrated in the background section as your topic is about the relationship between work and home, indicating that work roles and family roles will positively and negatively affect each other.

2.     “Teleworking parents often have to share their workspace, as schools have simultaneously instituted remote teaching. The need to combine telework with childcare in a frequently insufficient physical space and use frequently insufficient equipment has become burdensome for many parents. This is especially the case when both parents work remotely and have more than one child.” Any related references to support these statements?

3.     Please combine these two paragraphs into one paragraph “Thus far, telework during the pandemic has been variously described as a factor…reported lower stress during the telework days [27]” and “Studies also show that using digital technology at work seems to generate an…to lower work effectiveness [29].”?

4.     Whether one sentence, “It can be assumed that telework may play a significant role in the home-work and work-home relationships [36].”, can be one paragraph.

5.     The research niche seems to be not explicit. Why did you need to conduct this study? What is the significance?

6.     “The aim of the current study was to examine…” This paragraph should have a new sub-heading, like Aims.

7.     What kinds of organizations were invited to assist in survey distribution? Why would you choose these organisations? Based on what reasons or some sampling methods?

8.     In the discussion, how about working fathers? Whether fathers also have the same problems. Or this issue can be explored in future studies?

9. Please do not use point form in the conclusion. 

10.     Why the conclusion will be in front of Strengths, limitations, and future research? Please reorganise it.

Author Response

(The authors gave the same response as above.)

Round 2

Reviewer 1 Report

Dear Author

I read the paper, and am pretty convinced that this paper can be acceptable now.

Thank you.

Reviewer 3 Report

No. Thank you.